# Principled Decision-Making Workflow with Hierarchical Bayesian Models of High-Throughput Dose-Response Measurements

**DOI:** 10.3390/e23060727

**Published:** 2021-06-08

**Authors:** Eric J. Ma, Arkadij Kummer

**Affiliations:** 1Independent Researcher, Cambridge, MA 02139, USA; 2Department of Biosystems Science and Engineering, ETH Zurich, Mattenstrasse 26, 4058 Basel, Switzerland; arkadij.kummer@gmail.com

**Keywords:** hierarchical modelling, bayesian statistics, probabilistic programming, high-throughput measurements

## Abstract

We present a case study applying hierarchical Bayesian estimation on high-throughput protein melting-point data measured across the tree of life. We show that the model is able to impute reasonable melting temperatures even in the face of unreasonably noisy data. Additionally, we demonstrate how to use the variance in melting-temperature posterior-distribution estimates to enable principled decision-making in common high-throughput measurement tasks, and contrast the decision-making workflow against simple maximum-likelihood curve-fitting. We conclude with a discussion of the relative merits of each workflow.

## 1. Introduction

High-throughput measurements are a staple of biological measurements. A wealth of literature exists for the statistical analysis of high-throughput measurement data [1,2,3]. However, to the best of our knowledge, the application of hierarchical Bayesian models in high-throughput assay measurements is not widespread, with only a countably small number of papers leveraging hierarchical Bayesian methods [4,5]. However, the advantages of hierarchical models for estimation are well-known. For example, in [6], baseball players’ performance estimates were regularized away from extreme values. Players with fewer replicate observations of their fielding statistics had estimates shrunk closer to the population mean, though as more replicate measurements are obtained, the regularization effect diminishes. Other examples include its application in functional magnetic resonance imaging (fMRI) [7], survival analysis [8], genomic population analysis [9] and infectious agent risk analysis [10]. The main advantage of hierarchical models is that they allow for pooling of information between samples in a principled fashion, while still allowing for individual differences to show up, provided there is sufficient evidence for these differences. This property of hierarchical models may be desirable in a high-throughput measurement setting, providing a guardrail against being fooled by apparently desirable extreme measurements generated at random, or worse, through systematic error. Because hierarchical Bayesian estimation models require explicit, hand-crafted distributional assumptions, they also help us avoid canned statistical tests where our data might not necessarily fit the test’s assumptions (e.g., the *t*-test) [11]. We thus see a gap in the *application* of hierarchical Bayesian estimation in high-throughput biological assay measurements.

Recently, a global protein ’meltome’ was published, in which 41,730 proteins from across 16 species of life had their protein melting-points measured in a high-throughput fashion, spanning 1,114,710 data points that were released publicly [12]. Protein-melting occurs when a protein unfolds under thermal stress; a protein’s melting-point is thus of interest to biological research. In ecological studies, protein melting-points can reveal their host organisms’ properties (such as its probable characteristic temperature range). For biomedical applications, having a high melting temperature (i.e., high stability) while maintaining activity is a desirable property in protein therapeutics, and this property relates to a protein’s stability, which can be crucial for preserving a therapeutic protein until it is used.

The characteristics of these data are such that they very closely mimic the kind of measurement data generated in biological screening experiments used for drug hunting and protein engineering. Firstly, it is of the dose–response curve form; the melting temperature, being the mid-point of the curve, is analogous to the IC50 concentration values generated in chemical and enzyme screens. Secondly, it is measured in a high-throughput fashion, with extremely high numbers of samples measured, and with every sample (here, a protein from an organism) being treated equally in bulk. Importantly, they only have a single replicate measurement on which inferences, and hence decisions, are to be made. This means that measurements are effectively *sparse* with respect to a sample; even though there may be anywhere from 6–10 measurements of a sample, these are not biological replicate measurements of the same sample. This very closely mirrors what we observe in high-throughput screen measurements in drug discovery and high-throughput protein engineering, where most samples are measured once, and prioritization decisions have to be made on the basis of these measurements. As such, we reasoned that we could use this public data to illustrate how hierarchical Bayesian models can be used to guide decision-making under uncertainty in a high-throughput measurement setting. Where appropriate, we will contrast the Bayesian workflow against the traditionally used “separate curve-fitting” with maximum likelihood estimation. (As we are focusing on the application of hierarchical Bayesian methods to high-throughput measurements, we will only be discussing the relevant protein biochemistry in light detail. Any biological assumptions and conclusions we draw are held lightly; specific improvements to the model are discussed only in brief.)

This paper is organized according to the following short sections:An overview of the model and data to orient the reader (Section 2).Steps taken to validate correctness of the hierarchical Bayesian model (Section 3).An outline of how Bayesian posteriors can be used for principled decisions (Section 4).Further discussion of the advantages of hierarchical models, as well as limitations of this specific implementation (Section 5).Concluding thoughts on the promise of hierarchical Bayesian estimation in high-throughput assays (Section 6).

## 2. Model and Data Overview

### 2.1. Base Model Definition

The model provided in [12] for the melting temperature of a protein, as done by their measurement technique, is given by
$$f\left(L,a,b,T\right)=\frac{1-L}{1+e^{-\left(\frac{a}{T}-b\right)}}+L.$$

Here, *L* is the lower bound of the melting curve, which is treated as a random variable to estimate; $\frac{a}{b}$ is the characteristic melting-point of a protein, and *T* is the temperature at which a measurement is taken.

The modifications we made to the model to enable hierarchical modelling are detailed in the Materials and Methods (Section 7).

### 2.2. Data Characteristics and Summary Statistics

The data that were provided by the authors are 1.1 million rows of measurements, which contain:The species and extraction method that a protein was isolated from;The protein ID (a unique identifier encompassing its gene name);The gene from which the protein was expressed;The temperature at which an observation was taken;The fold change of the detected stable protein at that temperature, relative to the level at the lowest measured temperature.

Of the 41,730 proteins that were measured, 11,142 of them had no melting-point assigned. This is a direct result of the curve-fitting protocol used in the original analysis [12], which included criteria for data quality checks.

## 3. Validation of Model Implementation

### 3.1. Melting Curves

To validate whether our model implementation was done correctly or not, we employed a few diagnostics. Firstly, we rank-ordered proteins by their melting temperatures and spot-checked a subset of curves. Proteins that had low posterior uncertainty generally had high-quality curves, such as the blue curves in Figure 1. In this regime, the estimated melting temperatures from separate curve-fitting (dotted lines) generally agreed with the hierarchical Bayesian fits (solid line and shaded area), with minimal differences in the estimated melting temperature. As the posterior uncertainty increased, we gradually observed lower-quality curves (yellow and red in Figure 1. In particular, the red curves came from data that failed quality checks in the original analysis method, and hence did not have an assigned melting temperature. For the proteins that did have a melting temperature assigned, the magnitude of the difference between the hierarchical Bayesian estimate and the maximum likelihood estimate co-varied with the magnitude of the posterior distribution standard deviation, but was generally centered around zero (Figure 2).

### 3.2. Posterior Variance

In the meltome paper, proteins that had melting curves that failed quality control checks were not assigned a melting temperature. In a hierarchical Bayesian setting, we expect that these proteins’ curves should give us high uncertainty in their posterior estimates. Indeed, in our model fits, we observe this phenomena. Where our model imputed a protein’s melting temperature because of bad curve data (as assessed by the curve quality control checks), the posterior uncertainties for those melting temperatures were, quite reasonably, much higher than those without imputation (Figure 3).

## 4. Principled Bayesian Decision-Making in High-Throughput Settings

In this section, we discuss what hierarchical Bayesian modeling enables when used in lieu of separate curve-fitting.

In a setting where we measure high-throughput dose–response data, the goal is to find extreme-valued entities on the desirable side, such as low IC50 molecules for protein binders or high-stability proteins. Some downstream questions that need to be answered generally fall into the categories of:Which samples need higher-quality confirmatory measurements?Which samples should we take forward for further investigation in other measurement modes?

A classic constraint we find ourselves in is that of capacity: it is infeasible to take everything that is desirable. Additionally, it is desirable for us to have a ranking principle that factors in the confidence we have in any particular sample measurement. Leveraging the uncertainty in our estimates gives us a path towards principled decision-making.

In the next two sections, we outline how to address the decision-making dilemma in a principled fashion.

### 4.1. Acquiring Informative Measurements

One of our goals might be to acquire re-measurements of samples to improve the quality of our data set. To do so, we could rank-order samples by their posterior uncertainty in their measurements. (Figure 4a) In our example, this would be samples that have the highest posterior variance in melting temperatures; in classic molecular screening settings, this may be samples with the highest IC50 measurement uncertainties. Doing so would allow us to improve our data-set quality in an iterative fashion, with uncertainty being the guiding principle for re-measurement.

### 4.2. Confirming Optimal Measurements

In a vein similar to the acquisition of more informative measurements, we may wish to acquire confirmatory or secondary measurements on samples that have measurement values greater than a particular threshold. An example where this shows up is in enzyme engineering. Our primary assay measurement may be cheap to conduct, but the confirmatory assay may be more expensive. In addition, we may be doing multi-objective optimization, and the secondary properties that we are optimizing for may be similarly expensive to measure. Made concrete, enzyme thermal stability is often a cheap and high-throughput measurement to acquire, while enzyme chiral activity requires more sophisticated instrumentation, and is hence more difficult to conduct. In scenarios like this, our desired selection of samples are the ones that have the highest probability of being above some threshold we define *a priori* (Figure 4b). Here, access to the posterior distribution allows us to calculate the probability of a sample being greater than a particular value, and thus rank-order all samples according to this principle.

### 4.3. Prioritizing Samples for Further Modification

One other goal we might have would be to select samples from the pool of measurements as a baseline for further optimization. This is a classic protein and molecular engineering problem. Given the uncertainties in our measurements, by what principle could we select samples as our baseline?

This problem is similar to the previous section, where the desired outcome is the selection of the *best* new starting point. The difference lies in that in molecular and protein engineering, we are effectively optimizing, or searching, for samples that have *extreme* values. Higher activities, greater binding affinities, or smaller catalysis rates are what we are in search of. Hence, we once again desire samples that have the highest probability of being better than the rest as our new starting point. To calculate this, we would simply compare each sample’s posterior estimates and calculate the probability of superiority w.r.t. other samples’ estimates (Figure 4c). By contrast, under a classical setting, picking the sample with the most desirable point-estimate would only give us samples that have the highest expected value, which does not help us explore extremities as effectively.

## 5. Discussion

### 5.1. Hierarchical Bayesian Methods Enable Reasonable Estimates Where Separate Curve-Fitting Fails to Provide One

The key analysis “safeguards” that the meltome authors used to call a non-melter, and hence assign no melting temperature to a protein, was a combination of a threshold value for the lower-bound and a normalized area under the curve value [12,13]. However, one may ask the question: do we expect proteins to “not melt” under increasing thermal stress? Using a hierarchical Bayesian model, we effectively express a priori in the model structure that all proteins do *eventually* “melt” under thermal stress; proteins with low-quality measurements can still be assigned a melting temperature. As shown by our imputation distributions, the imputed values generally fall within the regime of the other proteins originating from the same organism. This stands in contrast to the original statistical analysis protocol, in which this model-wise structural assumption cannot be encoded in the model, thus requiring external heuristics as a safeguard for the “quality” of data. As for the question of which assumption is more reasonable, the answer would require debate and justification.

In the meltome atlas authors’ statistical methods, data that did not fulfill quality control criteria were given a null value for their estimated melting temperatures. By contrast, our use of a hierarchical Bayesian model that baked in biologically-relevant priors enabled us to provide imputed melting-points that fell within the general regime of those that *did* have melting-points estimated (Table 1). Because we had the posterior distributions inferred, further statistical analysis can leverage this model-based multiple imputation [14]. In doing so, we have managed to preserve hard-won data points without discarding them, even if their quality was questionable, and we have a framework for deciding which to re-measure if we so desired.

A natural caveat to this imputation method is that the imputed values are only as good as the model’s assumptions. For example, we do not factor in horizontal gene transfer or prior knowledge of a protein’s known function, both of which might affect our estimates of that protein’s melting-point. We reiterate, as a reminder to the reader, that where the data quality is good, separate curve-fitting and hierarchical Bayesian curve-fitting will generally match up; one of the value propositions of hierarchical Bayesian curve-fitting lies in *principled model-based imputation*, where the data quality might *not* permit this.

### 5.2. Limitations of Our Model and Inferential Procedure

We used approximate Bayesian estimation through ADVI [15] because of the large number of data points present. Though we ran it for a long number of steps (Materials and Methods), and could visually inspect for convergence loss, we could still be under-fitting. One other consequence of using ADVI is that the posterior distributions will have an under-estimated variance [16]. Suppose we are concerned mostly with the expected values of our posterior distributions. In that case, we would expect to run into a few issues. By contrast, if we are concerned with leveraging the posterior distribution for input design problems [17], such as optimizing melting temperatures, where the extremities of the posterior distributions are important, then the use of ADVI would give us less extreme extremities.

For this particular model, further hierarchical structures could have been imposed on protein measurements that came from both cells and lysate, or from two tissues. For simplicity’s sake, we treated these measurements as independent. However, incorporating this knowledge into the structure of the model would be an obvious next step to improve it for this data set. By extension, similar knowledge may be incorporated into other estimation models for different experiments. However, we do not anticipate that this point interferes with the main point of our article, which describes a decision-making protocol in high-throughput experimentation that leverages hierarchical modelling’s advantages.

## 6. Conclusions: The Promise of Hierarchical Bayesian Models in High-Throughput Biological Measurements

High-throughput biological measurements are known to be extremely noisy. Small sample volumes, large numbers of samples, and measurement of non-control samples with single replicates all contribute to the noise. The corollary is that with these, we may expect extreme-valued measurements to show up by pure random chance and systematic error. Even though in some settings, we may wish to find extreme values, we desire not to be fooled by random and systematic error. With shrinkage and posterior analysis, hierarchical Bayesian modelling provides us with a principled way out.

In our case study, we have provided an example where in curve-fitting scenarios, such as dose–response curves, regularization provided by hierarchical Bayesian models can act as a model-based safeguard against random extreme and noisy measurements. As we show in the case study, using a hierarchical model allows us to provide a curve parameter estimate *despite* noisy measurements, while providing posterior uncertainties as a natural and readily-available quality control measure. We have also described how these posterior uncertainties can help inform downstream experiments, such as deciding whether to re-measure a sample or a group of samples, or deciding which samples to take forward with regard to other measurements. As such, we believe that developing hierarchical models in high-throughput biological measurements may better leverage all available data and guide better iterative experimental design.

## 7. Materials and Methods

### 7.1. Hierarchical Bayesian Estimation Model

Here, we describe the model structure. A graphical representation of the model is available in Figure 5.

Our observed data, the fold change *y* across all proteins, are assumed to be Gaussian-distributed, with the central tendency μ for each observation given by the melting curve function *f* (which is a function of both temperature and the *L*, *a* and *b* parameters), and errors assumed to be homoskedastic.
$$y∼\mathrm{Normal}(f\left(L_{p,r},a_{p,r},b_{p,r},T\right),\sigma_{p})$$

Now, we describe the prior distribution assumptions that lead to each of the parameters.

*L*, the lower-bound, is given by a hierarchical prior, and is indexed on a per-protein basis, $L_{p}$, where p∈{0,1,2,…} indexes each protein. It is modeled by an Exponential distribution, with the population rate prior $L_{e}$ also given by a relatively flat Exponential distribution. (Subscript *e* indicates *experiment*.) This parameter helps us capture a *global* measurement lower-bound for each experiment, as we know from the experiment description [12] that the experimental conditions primarily influence the lower-bound.
$$L_{e}∼\mathrm{Exponential}\left(\frac{1}{30}\right)$$
$$L_{p}∼\mathrm{Exponential}\left(L_{e}\right)$$

Together, *a* and *b* give us the melting-point $T_{m}=\frac{a}{b}$, and are indexed on a per-protein basis. However, to model the organism-wide melting temperature, we introduce a per-organism $a_{r}$ and $b_{r}$, which are random variables modeled by a positive-only Normal distribution. (The subscript *r* is used instead of *o* for visual clarity, and *r* is the second letter of “organisms”.)

Given the curve equation, *a* and *b* have to be positive for the curve to take on a denaturation shape (i.e., decreasing *y* as temperature increases.) We then model a shift in *a* and *b* that is indexed by protein $\delta_{a,p}$ and $\delta_{b,p}$, which shifts the value of *a* and *b* from the organism level to give us a per-protein $a_{p,r}$ and $b_{p,r}$.

Additionally, $a_{r}$ and $b_{r}$ are given population priors ($a_{l}$ and $b_{l}$). (We use subscript *l* to denote that these are global priors across the tree of life.) Taken together, this model structure expresses that proteins from one organism most likely shares an underlying melting temperature distribution with other proteins from the same organism.

In mathematical notation, starting with the random variable *a*:$$a_{l}∼\mathrm{Normal}\left(500,1\right)$$
$$a_{r}∼\mathrm{BoundNormal}\left(a_{l},10\right)$$
$$\delta_{a,p}∼\mathrm{Normal}\left(0,3\right)$$
$$a_{p}=a_{r}+\delta_{a,p}$$

And now for the random variable *b*:$$b_{l}∼\mathrm{Normal}\left(10,1\right)$$
$$b_{r}∼\mathrm{BoundNormal}\left(b_{l},1\right)$$
$$\delta_{b,p}∼\mathrm{Normal}\left(0,1\right)$$
$$b_{p}=b_{r}+\delta_{b,p}$$

Their noise is assumed to be homoskedastic on a per-protein basis, i.e., $\sigma_{p}$, and is modeled with a standard HalfCauchy prior (with β=1).
$$\sigma_{p}∼\mathrm{HalfCauchy}\left(1\right)$$

We caution that this model is bespoke for the meltome dataset; other datasets with different structural assumptions will require a different model.

The model was implemented in PyMC3 [18]. Because of the size of the data set, we used ADVI for 200,000 steps with default settings instead of the default NUTS sampling, yielding an approximation to the posterior distribution from which we drew 2000 samples.

### 7.2. High-Throughput Measurement Data

High-throughput measurement data were sourced from the Meltome Atlas’ public-facing web server (at http://meltomeatlas.proteomics.wzw.tum.de:5003/) on 15 April 2020.

### 7.3. Separate Curve-Fitting

Separate curve-fitting was performed using the “curve fit” function in the “optimize” submodule of the SciPy library [19].

### 7.4. Posterior Curves

As the work was done prior to the availability of PyMC3’s posterior predictive sampling capabilities, the posterior curves were generated by passing the parameters’ posterior samples through the dose-response link function, thereby generating a distribution over dose-response curves for each sample. With PyMC3’s posterior predictive sampling capabilities, though, this is now a trivial function call.

## Figures and Tables

**Figure 1 entropy-23-00727-f001:**
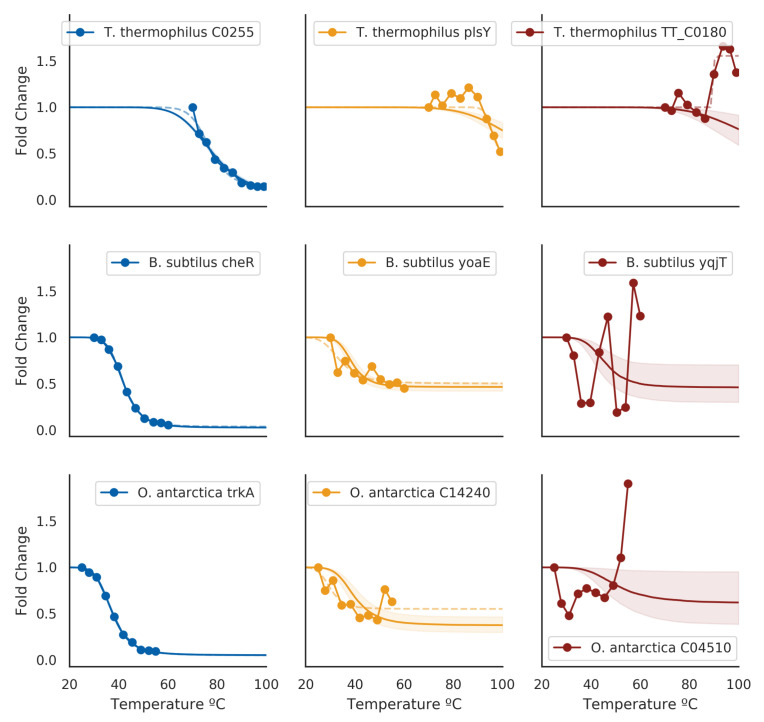
Example estimated melting curves against original measurement data for three species (by row). Blue figures are curves from proteins that had the lowest variance in estimated melting temperatures for each species. Yellow figures are curves from proteins for which melting-points are not obvious from the data and did not have an assigned melting temperature, but nonetheless could plausibly be assigned one. Red figures are curves from proteins that exhibited the highest variance in the estimated melting temperature for each species. Dotted lines indicate separate curve fittings using SciPy’s curve-fitting facilities (Materials and Methods); cases where errors were raised in curve-fittings are omitted.

**Figure 2 entropy-23-00727-f002:**
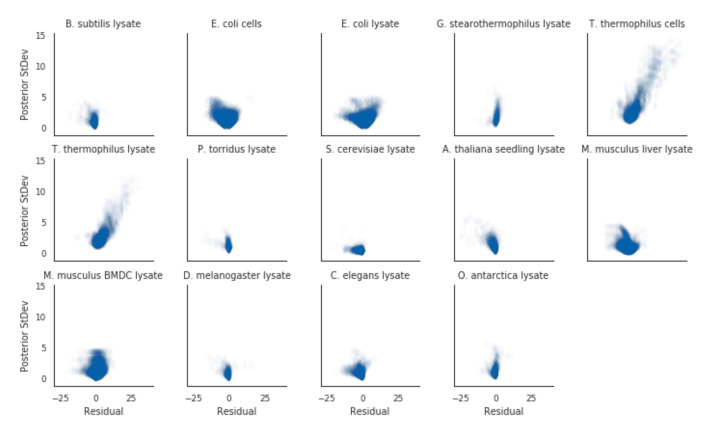
Proteins with greater discrepancies between the two methods for their estimated melting-points also had greater uncertainties for their Bayesian estimated melting-points. Residuals were calculated by taking the Bayesian estimated melting-point minus the separate curve estimated melting-point.

**Figure 3 entropy-23-00727-f003:**
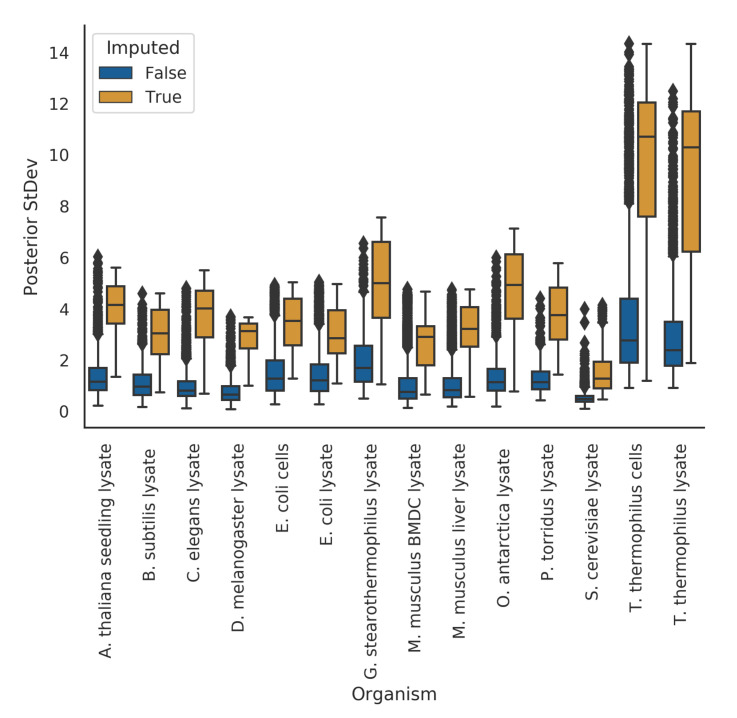
Imputed melting temperatures have higher uncertainty than non-imputed melting temperatures.

**Figure 4 entropy-23-00727-f004:**
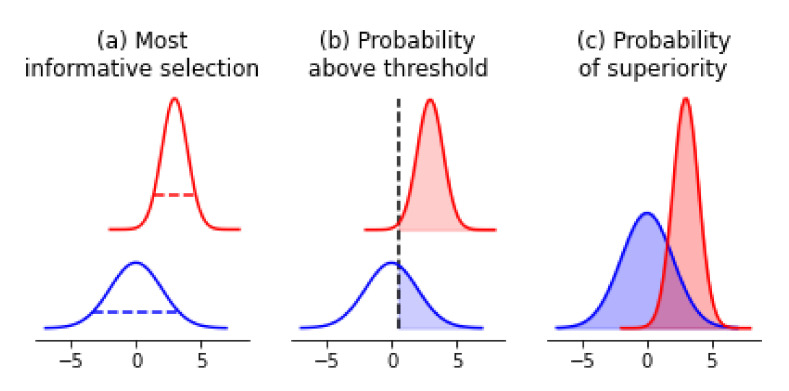
Probabilistic decision-making framework leveraging posterior distributions. (**a**) In choosing the next most informative re-measurement, we would suggest taking the blue sample because it has the highest uncertainty. (**b**) In choosing samples for confirmatory measurements that are above a threshold value defined *a priori*, we would suggest taking the red sample because it has the highest probability of being greater than a threshold value. (**c**) To decide which samples to use as a base for further modification towards extreme values, we would calculate the probability of superiority between all pairs of samples and identify the one that has the highest probability.

**Figure 5 entropy-23-00727-f005:**
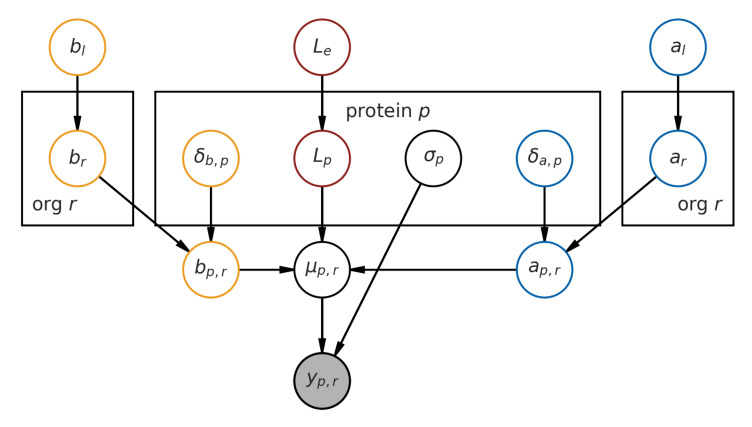
Graphical representation of the hierarchical model.

**Table 1 entropy-23-00727-t001:** Summary statistics of the imputed melting temperatures, excluding *D. rerio*.

Run Name	Mean	StDev	Min	25%	50%	75%	Max
A. thaliana seedling lysate	43.8	1.5	34.6	43.1	43.9	44.5	49.6
B. subtilis lysate	43.7	3.0	36.8	41.9	43.8	44.8	58.4
C. elegans lysate	44.0	3.5	34.2	42.2	44.5	45.4	57.6
D. melanogaster lysate	43.3	2.6	39.2	41.9	42.5	43.7	54.6
E. coli cells	54.1	3.5	45.4	51.8	53.8	55.7	67.1
E. coli lysate	55.2	4.6	45.9	51.8	54.1	57.9	67.3
G. stearothermophilus lysate	81.9	5.7	59.7	77.7	81.3	85.8	97.3
M. musculus BMDC lysate	49.4	2.0	44.0	48.2	49.4	50.7	60.0
M. musculus liver lysate	51.0	2.2	44.4	49.7	51.0	51.8	64.1
O. antarctica lysate	48.8	4.5	36.5	46.0	47.5	51.3	63.7
P. torridus lysate	72.9	3.6	65.2	70.5	72.2	74.5	83.6
S. cerevisiae lysate	47.1	2.3	40.9	45.7	46.8	48.4	55.4
T. thermophilus cells	108.5	8.5	80.7	104.8	110.3	114.5	125.0
T. thermophilus lysate	107.3	8.4	79.2	101.8	109.0	113.4	125.0

## Data Availability

Data were obtained from http://meltomeatlas.proteomics.wzw.tum.de:5003/ (accessed on 15 April 2021).

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
