# Peer review of "Principled Decision-Making Workflow with Hierarchical Bayesian Models of High-Throughput Dose-Response Measurements"

_entropy, 2021, doi:10.3390/e23060727_

Round 1

Reviewer 1 Report

This paper deals with the use of a hierarchical Bayesian model for on high throughput protein melting point data measured across the tree of life.  The paper contribution can be located in the demonstration of using the variance in melting temperature posterior distribution estimates to enable principled decision-making in common high throughput measurement tasks, and contrast the decision-making workflow against simple maximum-likelihood curve fitting.

The paper is well written and easy to read and understand. The choice of a Bayesian model is judicious but assumes a good knowledge of the apriori and posterior probabilities.

The results show the effectiveness of the proposed approach.

The paper can be improved on the following points:

  • Add a conclusion after the discussion that highlights the contribution of the paper.

  • Posterior probabilities are often unknown. How the authors proceed to build them.

  • Add to the conclusion of recent papers which deals with the problems of estimation and prediction of time series with probabilistic and deterministic methods applied in other fields such as industry. As for example: "Review of Health Indices extraction and Trend Modeling methods for Remaining Useful Life Estimation." Book Chapter Springer Nature Switzerland AG 2020.  Adaptive time series prediction and recommendation. Information Processing and Management. 2021

Author Response

We thank the reviewer for their thoughtful comments. Our responses are detailed below.

This paper deals with the use of a hierarchical Bayesian model for on high throughput protein melting point data measured across the tree of life.  The paper contribution can be located in the demonstration of using the variance in melting temperature posterior distribution estimates to enable principled decision-making in common high throughput measurement tasks, and contrast the decision-making workflow against simple maximum-likelihood curve fitting.

The paper is well written and easy to read and understand. The choice of a Bayesian model is judicious but assumes a good knowledge of the apriori and posterior probabilities.

The results show the effectiveness of the proposed approach.

The paper can be improved on the following points:

  • Add a conclusion after the discussion that highlights the contribution of the paper.

We agree that the paper could have a more explicit conclusion. In the original, the section beginning with "The promise of hierarchical Bayesian models..." is what we consider to be the conclusion of the paper. We have made that more explicit by promoting it to a section (rather than leave it as a subsection). Factoring another reviewer's comments, we have also provided an explicit outline to the paper earlier on to make the structure of the paper easier to follow. Please see the attachment.

  • Posterior probabilities are often unknown. How the authors proceed to build them.

We are a bit unsure as to what our reviewer is asking here, and would like to ask for some clarification. From what we know, posterior probability densities fall out naturally from Bayes' rule; in our case study, we use ADVI as a way of approximating the analytical posterior density, from which we draw computational samples. If this is what the reviewer is requesting we describe, then we have this present in the manuscript in the Materials and Methods section. However, if this is not what is being requested, we would be open to hearing more. 

  • Add to the conclusion of recent papers which deals with the problems of estimation and prediction of time series with probabilistic and deterministic methods applied in other fields such as industry. As for example: "Review of Health Indices extraction and Trend Modeling methods for Remaining Useful Life Estimation." Book Chapter Springer Nature Switzerland AG 2020.  Adaptive time series prediction and recommendation. Information Processing and Management. 2021

Though we are happy to cite other work to highlight connections between what we have done and what others have done, we are a bit unsure as to what the purpose of citing other work in the estimation and prediction of time series data would be. Would the reviewer be kind enough to help us understand the connection being proposed?

Reviewer 2 Report

The goals of this paper are given soon in the Introduction. There is a scarcity of papers producing applications of hierarchical Bayesian models in high throughput assay measurements, despite their advantages for estimation.

In this work the authors refer to a recent achievement [8] to produce an interesting case study that concerns a global protein ’meltome’. They have used this public dataset to illustrate how hierarchical Bayesian models can be used to guide  decision-making under uncertainty in a high throughput measurement setting.

Their starting point is the realization that the characteristics of these data (about the protein melting points of 41,730 proteins), that were measured in a high throughput fashion, resemble the kind of measurement data generated in biological screening experiments used for drug hunting and protein engineering. Remarkably, the measurements are effectively sparse with respect to a sample.

Sections 1.1 and 1.2 give the definition of the basic model, and the description of the characteristics of the data, respectively. Section 2 validates whether the model implementation is correctly done. In section 3 the authors argue why hierarchical Bayesian modeling enables them to achieve as an alternative to separate curve fitting. Section 4 gives a competent discussion, inclusive of the limitations of the model.

The article is professionally written. The overview of related literature is concise but sufficient.

There are graphical representations which help the reader to comprehend the workflows in empirical applications. Therefore in conclusion, I believe that this article is a very good fit for Entropy.

Minor comments:

As I said, the review of literature is very focused. For example, although the authors explain that hierarchical Bayesian  models are good for estimation, only one reference from 2009 is given. Can we see more evidences? Or any more recent advancement?

There is no outline of the paper. The distribution of material is confusing without a guide to the reader: the “Materials and methods” are given in the last Section 5, after the discussion. I am aware that this is hinted at the end of section 1.1; maybe a cross-reference to Section 5 rather than just its name, would be more informative.

Typos:

Line 12: a dot is missing after [1-3]

Line 22: they also help.

Author Response

We thank the reviewer for their thoughtful and thorough comments. In particular, we are heartened that the reviewer grasped the main point of the paper. Our responses to the review are listed below.

----------

Minor comments:

As I said, the review of literature is very focused. For example, although the authors explain that hierarchical Bayesian  models are good for estimation, only one reference from 2009 is given. Can we see more evidences? Or any more recent advancement?

We agree that the literature review has been quite light and was in need of additional citations to bolster the idea that hierarchical Bayesian models have been successfully used. We have added additional examples of where hierarchical Bayesian models have been applied to problems successfully.

There is no outline of the paper. The distribution of material is confusing without a guide to the reader: the “Materials and methods” are given in the last Section 5, after the discussion. I am aware that this is hinted at the end of section 1.1; maybe a cross-reference to Section 5 rather than just its name, would be more informative.

We agree that the structure could have been made a bit clearer. To help the reader, we have added in a short outline at the end of the introduction to orient the reader a bit better. Factoring in another reviewer's comments, we have also done a minor restructure of the paper such that the conclusion is made a bit more explicit.

Typos:

Line 12: a dot is missing after [1-3]

Line 22: they also help.

Thank you for catching these. We have updated the manuscript. Please see the attachment.
